# Comparison of Vocalization Patterns in Piglets Which Were Crushed to Those Which Underwent Human Restraint

**DOI:** 10.3390/ani8080138

**Published:** 2018-08-08

**Authors:** Nichole M. Chapel, Jeffrey R. Lucas, Scott Radcliffe, Kara R. Stewart, Donald C. Lay

**Affiliations:** 1Department of Animal Sciences, Purdue University, West Lafayette, IN 47907, USA; jradclif@purdue.edu (S.R.); krstewart@purdue.edu (K.R.S.); 2Department of Biological Sciences, Purdue University, West Lafayette, IN 47907, USA; Jeffrey.r.lucas.1@purdue.edu; 3USDA-ARS, Livestock Behavior Research Unit, West Lafayette, IN 47907, USA; Don.Lay@ARS.USDA.GOV

**Keywords:** piglet crushing, swine vocalization, sow-piglet communication

## Abstract

**Simple Summary:**

Piglet crushing (the process by which a sow sits or lies on and crushes her piglet) is a welfare issue in pig husbandry. In order to understand why crushing occurs, and avoid hardship to a piglet, recordings of distressed piglets are often used to simulate crushing events to measure sows’ behavior. Unfortunately, it is not known if the call produced by the distressed piglet is similar to a piglet being crushed, making the evaluation of the results difficult. We recorded calls of piglets during crushing and compared the calls to those produced by restrained piglets. When crushed, piglets have a deeper call than piglets which are restrained by a human. Restrained piglets call as loudly as crushed piglets. In conclusion, sufficient differences exist between restrained and crushed piglets that restrained calls alone should not be used to understand the conditions in which a sow will respond to the distress calls of her piglets. Future research should include measuring sow behavior in response to Crushed and Restrained calls.

**Abstract:**

Though many studies focused on piglet crushing utilizing piglet vocalizations to test sow response, none have verified the properties of test vocalizations against actual crushing events. Ten sows were observed 48 h after parturition, and crushing events were recorded from all sows. When a crushing event occurred, a second piglet within the same litter was used to solicit a vocalization through manual restraint to compare restrained piglets’ call properties to those of crushed piglets’. A total of 659 Restrained calls and 631 Crushed calls were collected. Variables were gathered at the loudest point in a call, and as an average across the entire call. Crushed piglets had a lower fundamental frequency (*p* < 0.01; Crushed: 523.57 ± 210.6 Hz; Restrained: 1214.86 ± 203.2 Hz) and narrower bandwidth (*p* < 0.01; Crushed: 4897.01 ± 587.3 Hz; Restrained: 6674.99 ± 574.0 Hz) when analyzed at the loudest portion of a call. Overall, piglets which were crushed had a lower mean peak frequency than those which were restrained (*p* = 0.01; 1497.08 ± 239.4 Hz and 2566.12 ± 235.0 Hz, respectively). Future research should focus on measuring sow reactivity to Crushed and Restrained piglets to continue to improve research practices.

## 1. Introduction

Piglet crushing, the process where a sow sits or lies on a piglet, accounts for almost half of all preweaning deaths in swine production [1]. Although piglet crushing is a common occurrence on farms, little agreement can be made about the cause of piglet crushing by sows. Attempts have been made to understand if piglet crushing is related to tactile, visual, or auditory presence of a piglet, as different stimuli may affect piglet identification and sow response [2]. Piglet vocalizations, arguably the most commonly researched signal used in piglet crushing research, can be gathered during an actual crushing event; however, this is difficult, because crushing events are unpredictable [3]. To reduce animal use and distress, many researchers attempt to simulate a crushing event through physical manipulation of a piglet to induce vocalizations, and then perform a playback of the recorded piglet vocalization to measure sow response [4,5].

These methods are not ideal, as calls are recorded by different handling techniques and have not been validated against an actual crushing call to ensure the validity of the experimental results. Moreover, recorded calls solicited from piglets are gathered through restraint, not from an actual crushing event. Piglets can change their vocalization structure to communicate varying health conditions to a sow, as observed when pig handling changes from restraint to castration [6]. It stands to reason that pigs have unique calling principles to demonstrate extreme need or danger, similar to yells in human communication which have properties which are distinct from normal speech [7]. Differences may exist between a Restrained call and a Crushing call due to different levels of threat to a piglet’s well-being. These differences will result in decreased data quality and increased variability of results for piglet crushing research. Therefore, the objective of this study is to compare the call structures during piglet restraint and piglet crushing for use in validating piglet vocalizations for future crushing research, ultimately ensuring greater accuracy of data collection and results.

## 2. Materials and Methods

All procedures performed on sows and piglets were approved by the Purdue Animal Care and Use Committee (PACUC, #1607001446) and performed at the Purdue Animal Science Research and Education Center swine farm in West Lafayette, IN, USA.

### 2.1. Animals and Procedures

Gilts were excluded from this study, as previous crushing history was used for enrollment selection. A total of 10 Landrace × Yorkshire sows who had crushed at least 1 piglet to death in previous litters were enrolled in this study. Sows were housed in farrowing crates (0.61 m × 2.29 m) with finger bars. Heat lamps were provided in the same location in all crates. Sows were fed a total mixed ration in the morning at the same time each day. Piglets were not handled or processed until after completion of the study.

Video recordings were performed using infrared cameras (Nuvico CB-HD65N-L Bullet camera; Nuvico; Englewood, NJ, USA) which were placed prior to farrowing. Recording equipment was maintained until a crushing event occurred in the first 48 h post farrowing, as the majority of crushing events occur within the first two days of a piglet’s life [8]. An automatic recording device (SM3, Wildlife Acoustics, Maynard, MA, USA; hereafter referred to as ‘microphone’) was placed above the farrowing crate along the top and in the center of the crate, and maintained with the camera. The microphone was within 1 to 1.5 m of all crushing events. Due to the microphone’s location when a crushing occurred, crushing events were the loudest element recorded by the microphone, and therefore all call analyses were performed on the crushed piglet’s call. 

All sows crushed one piglet to death; therefore, calls were analyzed from 10 crushed piglets. Piglet crushes were identified by visual inspection of piglets, and if more than one crushing event occurred for a single sow, only the first crushing was analyzed. When a sow crushed a piglet (’Crushed’; Figure 1a), another piglet from the same litter was haphazardly selected and was carried to a room without any other pigs present. A distress call (’Restrained’; Figure 1b) was collected approximately 1 m from the microphone. The Restrained call test was performed by an unknown handler to the piglets, similar to other crushing simulation studies, as other studies do not specifically use a familiar caretaker for these tests. The Restrained call was collected by firmly holding the piglet around the ribcage and gently applying pressure with the handler’s hands to the piglet. If necessary, the piglet was also held on its side to induce a vocalization. Calls of restrained pigs were collected for approximately 30 s. If the piglet did not vocalize, a different piglet was chosen until calls were collected. A total of 10 piglets were recorded for Restrained calls.

Crushing events were matched between video and audio, and calls from each piglet were identified. Calls for a single event (either one piglet being crushed or one piglet being restrained) were placed into individual sound files for analysis using audio editing software (Adobe Audition, Adobe, San Jose, CA, USA). Audio files were saved as uncompressed.WAV files. Call analysis was performed using sound analysis software (Avisoft-SASLab Pro, Avisoft Bioacoustics, Glienicke, Germany). For each file in the sound analysis software, individual calls were identified and marked by a single, trained individual. Each individual call was then analyzed by the software for parameters outlined in Table 1 (Adapted from [5,9]). All parameters (except total duration and interval to peak) were analyzed using data from the loudest point in the call (hereafter referred to as ‘max’). Parameters were also averaged across the entire call (hereafter referred to as ‘mean’). Mean takes into account the entire call for each parameter, while max only analyzes the loudest segment in each call. This is the portion of the call most likely to be heard by the sow. Frequency parameters were calculated for frequencies presented above 30 dB to minimize potential sources of noise associated with the environment and not the piglet call. To analyze 1st and 2nd formants, calls were analyzed in PRAAT version 5.4.15 [10], and visual observation was used to confirm the location of each formant.

### 2.2. Statistics

Data were analyzed using an analysis of variance with a mixed model design in SAS 9.4 (SAS Inst. Inc., Cary, NC, USA). Piglet was the experimental unit. Repeated measures were used with treatment nested in sow to account for multiple calls produced per Crushed and Restrained piglet. Fixed effects included animal treatment (Crushed or Restrained) with sow as a random effect. Variables were assessed for normality of distribution, homogeneity of variance, and linearity of the data. Significance was defined as *p* < 0.05 and the least squares means ± the standard error is presented for all data.

## 3. Results

A total of 659 Restrained calls (65.9 ± 8.0 calls per piglet), and 631 Crushed calls (62.9 ± 17.1 calls per piglet) were collected from all 10 sows. The duration of calls produced by piglets was not different between Crush and Restrained (*p* = 0.33, Figure 2a). Start to peak in each call was similar between treatments (*p* = 0.49, Figure 2b). 

### 3.1. Max Parameters

All max vocalization data are summarized in Table 2. Overall, restrained piglets had higher frequencies in the calls compared to crushed piglets. When parameters were measured at the loudest point in each call, the peak frequency was higher by 1385.47 Hz in restrained piglets than crushed piglets. However, there were no changes in the amplitude of the calls when being restrained or crushed. Animal handling affected the minimum and maximum frequencies in each call, as restrained piglets had a higher minimum frequency and maximum frequency than crushed piglets. The bandwidth from minimum to maximum frequency was wider in restrained piglets than crushed piglets. The fundamental frequency was also higher in restrained piglets by 691.3 Hz. The 1st formant was higher in restrained piglets, however the 2nd formant did not vary between restrained and crushed animals. The frequencies found at the quartile 25%, 50%, and 75% were higher in restrained piglets than crushed piglets. Restrained piglets did not differ from crushed piglets when calls were measured for degree of entropy or harmonic-to-noise ratio.

### 3.2. Mean Parameters

Table 2 contains all mean parameter data, which can be summarized as follows. The mean peak frequency of the calls was higher by 1069 Hz in restrained piglets than crushed piglets. Overall, calls were not different in mean peak amplitude when averaged across the entire call. The mean minimum frequency was lower by 689 Hz in crushed piglets; however, the mean maximum frequency was not different between restrained and crushed piglets. Neither the mean bandwidth nor mean fundamental frequency were different between the two treatments. The mean 1st formant was higher in restrained pigs; however, the mean 2nd formant was not different between crushed and restrained piglets. The mean frequencies at the quartile 25% and 50% were higher in restrained piglets than crushed piglets, but the quartile 75% was not different. The entropy and harmonic-to-noise ratio were not different between crushed and restrained piglets.

## 4. Discussion

The aim of this study was to measure if calls produced by piglets which were crushed by a sow were different from calls produced during restraint by a human handler. As differences between these two procedures were found, these data are to be used in piglet crushing research to help future researchers develop better methodology for simulating piglet crushing by a sow.

In this study, piglets which were crushed by the sow had a lower peak frequency compared to those which were restrained by a human handler. Previous work investigating if piglets can vary calls with intensity of pain has yielded mixed results. Some have found that during castration, piglets have a lower peak frequency in their calls when they receive anesthesia prior to castration compared to those which are only restrained [11]. Additionally, the majority of call research during piglet castration has found that castration produces more and higher frequency calls than restraint [12,13]. Other processing procedures, such as tail docking, produce higher frequency calls compared to restraint, but ear notching has no effect [13]. Also, applying both ear notching and tail docking to piglets did not change call frequency compared to sham processing [14]. 

Frequency differences observed between crushed and restrained piglets could be related to interference with lung capacity of the piglet. During a crushing event, any part of a piglet’s body can be trapped underneath a sow. When the head or torso is trapped, it is likely that a piglet cannot breathe correctly due to broken ribs and the sow’s weight, therefore preventing the piglet from being able to vocalize adequately. During a restraint call, the piglet’s face is not muffled and it is able to breathe correctly. If playback research is used to simulate crushing, extra attention should be given to the frequencies within the playback call, as calls with lower frequencies may provide a better signal for research.

Also, the 1st formant was higher in restrained piglets than in crushed piglets, but the 2nd formant was not different between the two conditions. First and 2nd formants have previously been used in studies of swine, and are easily distinguishable for calculation [15]. Formant structures are critical for communication properties. In humans, formant location and structure develop the principles of sounds in the English language [16]. In pigs, body size affects sound structure, as the 1st and 2nd formants are inversely related to body size [15]. Though body size was not collected in this study, previous research shows that smaller piglets are more likely to be crushed by the sow [17]. This would suggest that crushed piglets should have a higher formant; however this was not observed. The results of this study may indicate that formant structure is disturbed during a crushing event by the sow’s body, which may be an important factor during crushing research utilizing a playback of a piglet’s call. As there were no differences between the 2nd formant for each treatment, pigs may have been relatively uniform in size, as is the goal in commercial practice. Also, work by Illmann and others [5] demonstrated that body weight of piglets had no effect on several frequency parameters during human restraint (such as minimum and maximum frequency, peak frequency, and quartiles).

No differences were observed in the duration of each call for crushed or restrained piglets. This agrees with previous findings, as others [9] have measured variation in squeals and grunts produced by piglets when piglets were restrained on their back, and found no differences in the duration of either call type. We also observed no difference in amplitude between crushed and restrained piglets. Piglets are loose in the crate during crushing, which makes it impossible to completely standardize the distance from the piglet to the microphone; however, we placed the microphone in the same location within a crate every time. Sows were under confinement which limits the maximum distance from the microphone to a crushed piglet. Others have reported no change in the amplitude of piglet vocalizations during handling or injections [18]. Amplitude could be affected by how the piglet is crushed by the sow, due to the weight of the sow and decreased lung capacity.

Piglets which were crushed did not vary in harmonic-to-noise ratio or entropy from those which underwent restraint by a human. In humans, yells contain unique harmonic-to-noise ratio and entropy properties when compared to talking [7]. Harmonic-to-noise ratio has been proven to be modulated in pigs for calls of varying intensity, with grunts having a lower harmonic-to-noise ratio than recordings of piglet squeals [9]. In our study, piglets which were crushed would likely experience increased intensity during a crushing than during restraining; however, piglets were consistently performing squeals during both conditions. Harmonic-to-noise ratio and entropy appear to have species specific properties, with differences between grunts and squeals for pigs [9], but not among squeals in different conditions.

## 5. Conclusions

Overall, these data demonstrate that calls made during manual restraint are not comparable to those produced during an actual crushing event. Crushed piglets have decreased peak and fundamental frequency, 1st and 2nd formant, bandwidth, and Q25, Q50, and Q75 frequencies. Future research should include using both crushing and restrained calls in playback research to measure the effect on sow reactivity to enhance the scientific field as we begin to understand sow-piglet communication. Also, with the rise in automated animal monitoring systems, continuing to identify specific components of calls produced during piglet crushing or extreme pain could be a useful addition to current monitoring. This technological prospect would help future producers to minimize distress to piglets, deliver timely euthanasia, and increase piglet livelihood.

## Figures and Tables

**Figure 1 animals-08-00138-f001:**
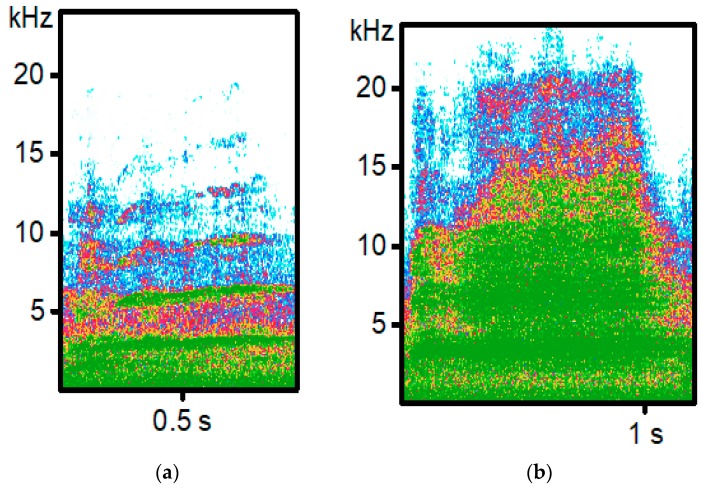
Representative spectrograms recorded from piglets in two different conditions. (**a**) Crushed by a sow; (**b**) Restrained by a human handler.

**Figure 2 animals-08-00138-f002:**
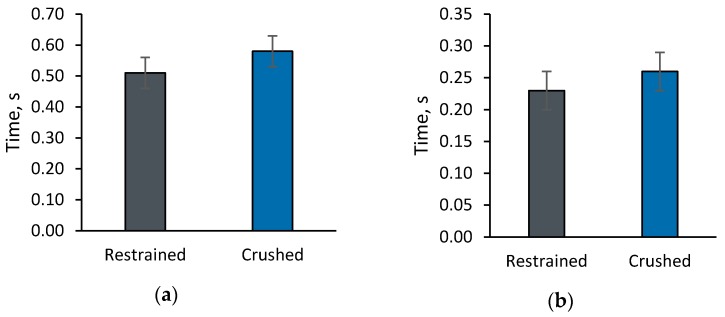
Time parameters of calls produced by piglets during manual restraint by a human handler (*n* = 10; 659 calls total) or during a crushing event by a sow (*n* = 10; 631 calls total) in the first 48 h of life. (**a**) Total duration of each call produced; (**b**) Start to peak within each call.

**Table 1 animals-08-00138-t001:** Parameters measured on piglet calls produced during manual restraint by a human or during crushing by sows during the first 48 h of life.

Parameter	Description
Total Duration; s	Total time from the onset to the end of a call
Start to peak; s	Distance from the beginning of the call until the loudest element is reached
Peak Frequency; Hz	Loudest frequency found within a call
Peak Amplitude; dB	Measurement of the highest energy within a call
Minimum frequency; Hz	The lowest frequency within a call above the threshold (30 dB)
Maximum frequency; Hz	The highest frequency within a call above the threshold (30 dB)
Bandwidth; Hz	The difference between the minimum and maximum frequency
Fundamental frequency, Hz	Lowest frequency in a distinct harmonic structure
First formant, Hz	First detectable concentration of frequencies forming a band structure within a call
Second formant, Hz	Second detectable concentration of frequencies forming a band structure within a call
Quartile 25%; Hz	Frequency found at 25% of the total energy in call
Quartile 50%; Hz	Frequency found at 50% of the total energy in call
Quartile 75%; Hz	Frequency found at 75% of the total energy in call
Entropy	The randomness within a call where zero is a pure-tone and one is completely random noise
Harmonic-to-noise ratio	Ratio of the degree of harmonic sound to additional noise produced within a call

**Table 2 animals-08-00138-t002:** Sound parameters measured on piglet calls collected during manual restraint by a human (*n* = 10; 659 calls total) or during crushing by a sow (*n* = 10; 631 calls total) during the first 48 h of life.

Parameter	Restrained (max)	Crushed (max)	*p*-Value	Restrained (mean)	Crushed (mean)	*p*-Value
Peak Frequency; Hz	2953.74 ± 335.3	1568.27 ± 343.8	0.01	2566.12 ± 235.0	1497.08 ± 239.4	0.01
Peak Amplitude; dB	85.88 ± 1.5	82.14 ± 1.6	0.11	76.59 ± 1.6	73.83 ± 1.6	0.24
Minimum frequency; Hz	972.74 ± 121.6	358.96 ± 127.8	0.001	1130.08 ± 155.0	441.27 ± 158.2	0.006
Maximum frequency; Hz	7619.69 ± 644.8	5293.22 ± 656.4	0.02	5759.86 ± 426.7	4392.23 ± 433.3	0.37
Bandwidth; Hz	6674.99 ± 574.0	4897.01 ± 587.3	0.04	4587.47 ± 334.5	3904.95 ± 342.1	0.17
Fundamental frequency, Hz	1214.86 ± 203.2	523.57 ± 210.6	0.03	965.96 ± 149.6	619.72 ± 151.0	0.12
First formant, Hz	2523.07 ± 118.2	1989.74 ± 121.0	0.005	2281.49 ± 98.3	1984.69 ± 100.	0.05
Second formant, Hz	3603.89 ± 85.5	3395.68± 87.9	0.11	3533.48 ± 70.6	3414.48 ± 72.0	0.25
Quartile 25%; Hz	2866.29 ± 279.0	1630.95 ± 285.4	0.006	2474.03 ± 196.8	1701.45 ± 199.7	0.01
Quartile 50%; Hz	3759.52 ± 322.5	2338.96 ± 328.4	0.006	3899.07 ± 258.9	2966.97 ± 262.1	0.02
Quartile 75%; Hz	5630.20 ± 436.1	3919.92 ± 442.7	0.01	5943.15 ± 362.1	4887.73 ± 365.8	0.06
Entropy ^1^	0.40 ± 0.1	0.36 ± 0.1	0.35	0.48 ± 0.1	0.46 ± 0.1	0.69
Harmonic-to-noise ratio	16.49 ± 2.5	19.21 ± 2.5	0.46	18.54 ± 2.8	20.07 ± 2.8	0.71

^1^ Refer to Table 1 for description.

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
