# Peer review of "Comparison of Vocalization Patterns in Piglets Which Were Crushed to Those Which Underwent Human Restraint"

_animals, 2018, doi:10.3390/ani8080138_

Round 1

Reviewer 1 Report

Review of manuscript Comparison of vocalization patterns in piglets which were crushed to those which underwent human restraint (animals-330653)

General remarks:

This manuscript presents a comparison of call characteristics of piglets that were either crushed by sows or restraint by humans. I think this an interesting area of study that to my knowledge has not been done before.

In general, the manuscript is clearly written and results clearly presented. However, I feel there is information lacking from the manuscript in regards to methodology and measurements taken to help readers with the interpretation of the findings. This should be addressed before being considered for publication.

General comments:

In general, more information should be provided for the methodology of the study. Right now several things are not completely clear to me making it difficult to properly interpret the manuscript. I believe there are many more relevant studies out there that can be referenced and used to strengthen the manuscript.

Information regarding sow/piglet housing (dimensions, safe guards etc.) and selection is missing. The justification of not including gilts mentioned crushing history was used but it is not explained how this was used or how sows were selected. Did you include sows who showed crushing in previous farrowing, >1 crushing in previous farrowing, crushing in all previous farrowings etc.? What was the parity of the sows and did you adjust for that if it differed? Did all 10 sows crush a piglet (or more)?

Similar, ‘haphazardly’ selecting a piglet for restraint is somewhat strange terminology. What about male / female piglets ratio, especially if you mention that castration affects vocalization or maybe male and female piglets in general differ in their vocalization patterns? On that note, what other procedure did the piglets undergo during the 48h post farrowing period. It is not clear to me if 1 piglet = 1 recording = 1 call or could 1 recording for 1 piglet exists of multiple calls? And if so, did you analyse these different calls or how does the software handle that? How does the software work in general, does it provide the values for all the parameters or do the researchers have to indicate certain regions within the call, make calculations etc.?

For the crushing recordings, these where of a period of 48h how did you identify which call was associated with the crushing? I assume it is recording all vocalizations of the sow, other piglets also outside of the crushing (maybe even the other pens) and that it is difficult to estimate when the crushing exactly occurred (if there is a time stamp on the recordings). What did you do if by chance more than 1 piglet was crushed (hopefully this didn’t happen J).

Please elaborate on how the restraint was performed e.g. was it always the same handler or different people? Maybe this is more a discussion point, but have you considered the human-animal relationship and how this could have influenced piglet vocalization during restraint?

My other main concern is with the actual parameters measured. Quite a lot were measured but I am not sure of the justification or reasoning why each is recorded (why for peak and mean?), or more specifically in the interpretation of each. The authors attempt to address this in the discussion but in my opinion it is a bit superficial. Now this could be in part due to my limited knowledge in this field, but if the authors want to reach a wide audience I would recommend to try and make this clearer in the manuscript.

More specific comments regarding the manuscript follow below.

Simple summary:

L16,18,19: The word ‘researchers’ is repeated quite frequently in this section. Please rephrase to make it flow better. Possible suggestion: ‘In order to understand why crushing occurs, recordings of distressed piglets are often used to simulate crushing events to measure sow’s behaviour’.

L20: ‘little is known if’. Rephrase e.g. ‘it is not known if’

 Abstract and keywords:

If possible, please add some information on software and analysis done, actual result values (e.g. lower peak frequency is mentioned but not clear how it compares in both groups), how many crushing/restrained calls etc.

General note: sometime crush(ed)/restrain(ed) is capitalized and other times not (e.g. L35 ‘crushed’ and L36 ‘Restrained’)

L31: ‘to compare call properties of a call to’ rephrase for clarity e.g. ‘to compare restrained piglets’ call properties to those of crushed piglets’.

L38-39: I expect to see some suggestions in the discussion on what should be done to simulate crushing if not only restrained calls can be used.

Introduction:

The introduction is concise and well written. I only have a few minor comments.

L43: ‘through which’ change to ‘where’

L45: ‘the farm’ change to ‘farms’

L48-51: Sentence is quite lengthy. I would rewrite it to ‘actual crushing event, however, this is difficult because crushing events are unpredictable.’ (Crushing events being unpredictable in my mind automatically imply you will need to record many sows in order to catch them as not all will actually crush piglets, or you might simply miss the event).

L58: Double check that there is only a single space at the start of the sentence ‘Piglets can change’.

Material and methods:

Please see my general comments for the more extensive questions I have regarding this section.

L76: Is there a specific reason the 48h post partum were recorded, is the most likely time period to see crushing events? If so you can support this in the manuscript with a reference to make it stronger.

L85: ‘at least 30 sec’ and maximum then? How many calls are recorded during this time, do you only analyse one? See also general comments.

L94: Similar to my previous comment, why where frequency parameters only calculated for frequencies above 30dB?

L95: ‘confirm location’ of what?

Regarding statistics, please consider based on my earlier comments whether the model needs to be improved by accounting some of the other factors.

L104: Tukey adjustment are not needed if you only compare 2 groups, they should be the exact same the real P-values.

L104: ‘means and SE’, I assume this is the LS means and its SE?

Results:

In general, the results are okay, however reading it is a bit repetitive. Try and rephrase it a bit to make it more lively/interesting for your readers. They are important findings! Readers just need a bit of a helping hand in navigating/understanding results. P-values should only be presented in table or text but not both.

L108: refer to figure 1A and 1B

L143: ‘Max’ should be ‘Peak’ according to authors stating that the loudest point in the call will be referred to as peak in L92.

L160: ‘peak amplitude’ should this not be ‘mean amplitude’? Might be a mix up or maybe it is technically ‘mean peak amplitude’ but it makes it a bit confusing with the peak / mean terminology introduced in the manuscript.

L164: ‘respectfully’ change to ‘respectively’.

Discussion

See the general comments to elaborate on the different parameters and what they mean to help readers interpret the study. I would suggest to very briefly repeat the aim of the study to keep your readers on track when going in the discussion.

One of the things that stood out to me in the discussion is that the authors introduce the idea that crushed piglets cannot vocalize the same way as restrained piglets when the head/torso is trapped which makes sense (do we know if that was the case for the crushed piglets in this study?) However, you are then contradiction this by saying that ‘piglets can modulate their calls during crushing’ (L184). If piglets cannot vocalize because they are trapped it is not that they are changing their call, the call is simply different because of their circumstances. It might be a simple matter of choosing different wording.

L180-183: This can be condensed; it is a bit saying the same thing twice. Also add the reference.

L191-198: This is a bit vague. Please rephrase and clarify/ elaborate. In my understanding: body size inversely related to formants à smaller piglets = larger formants à smaller piglets more likely to be crushed à suggests that crushed piglets would have larger formants?

L199: regarding duration of calls, in the restrained calls you recorded at least 30 sec. So technically, you are influencing the duration, if you have a call that lasted 60 seconds and you capped the recording at max 40 sec you will see a different value (40 sec duration) then if you had capped the recording at 90 sec (60 sec duration). See also previous comments.

L201-206: The authors introduce a new concept ‘arousal’ but it is not clearly explained or defined. I am missing the point of this addition to the manuscript (coming back in L211). Either clarify or delete.

Conclusions

L229-232: I do not see how these results can help producers optimize environment and barn design, I do not think producers will be able to identify or hear these specific call characteristics that are measured by software. I get where the authors are going but it is more a future implication that can be achieved by further research on crushing events with more accurate methods to simulate crushing etc. I think that is more a section that could be included in the discussion in addition to offer some of these other ways for researchers to simulate crushing if they cannot only use the restrained calls.

Figure 1

Provide n = for the number of calls that are included for the restrained and crushed groups.

L111: ‘by piglets’ delete. Redundant, it is the same for Fig. 1B.

Tables

Double check journal guidelines that formatting with all these row lines is correct.

Table 1

Please rephrase the title ‘collected during manual restraint’ and ‘were crushed by a sow’. This does not match (e.g. ‘during crushing by sows’). See also general comments (is the table adapted from somewhere if so add reference).

Table 2

Max parameters should be peak parameters (see previous comment). Change decimal points so that the mean is one decimal point less that the SE. Again, provide n = for number of calls within each group.

L112: ‘Sound parameters’ change to ‘Parameters’.

Reviewer 2 Report

This manuscript describes the difference in the vocalization of piglets either being crushed by the sow or being restraint by a human. The results show that the vocalization of a piglet that is being crushed is different from piglets that are being held and that therefore using piglet restraint is not a valid model for simulating crushing for research purposes. This is relevant for research in this area, which may increase as the industry/legislation is slowly moving in the direction of implementing free farrowing pens in which the occurrence of crushing is even higher.

The manuscript is well written and the few suggestions that I have are only an addition. The main thing that does need to be stated is the sample size. It is not clear whether the vocalization of one C and one R piglet per sow were taken, meaning that all sows at least crushed one, or that this number differs (e.g. no piglets or multiple piglets crushed per sow).

Specific comments:

line 16: please state pig husbandry instead of pork industry. When working with live animals they are still pigs and not yet pork.

line 17: remove 'many', this research method has been used but many is a bit an overstatement, certainly when looking at the number of references cited in relation to that.

line 44: this is the same, or even higher in other parts of the world and does not need to be specified that it is US only. Please remove mention of US or replace by a statement that is is a general problem across countries.

line 73: were sows selected based on previous history of crushing? 

line 76: does this automatic recording device include the mic? Please specify details on the mic if relevant.

Line 87: mention how many C and how many R piglet vocalizations were obtained, from how many sows.

Results: it would be nice if the vocalization could be more visualized, e.g. with spectrogram. In addition it would be even better to have an audio of both types of vocalizations in online material, either in the journal if this option exists or with a link to an online file. The difference in call is quite clear and it is informative for, for example, researchers that are not yet familiar to the difference in vocalization.

Line 181-182: is this reference 9 as well? If so then enter [9] at the end of the sentence for clarity.

Discussion: do take in mind that even though the piglet vocalization does differ, this does not give information on how the sow perceives the call. A possible addition to the current work would have been to record the sows' response to both vocalizations to see if the difference in call is perceived/relevant for the sow (which would biologically be so, but in practice doesn't  seem so straightforward). Just a suggestion and I leave it up to the authors whether they want to mention something along these lines in the discussion or not.    

Author Response

Please see attached word document.

Reviewer 3 Report

Dear Authors,

I have evaluated the manuscript “Comparison of vocalization patterns in piglets which were crushed to those which underwent human restraint” and I have doubts on its acceptability in the present form.

The aim of their research should be better clarified. Could this work be used as a refinement for future research? How could it contribute? Would it be possible to design “anti-crushing systems” using recorded vocalizations? Which are the Authors’ suggestions with this respect? The sentence in the conclusions (lines 229-232) gives a vague perception but no suggestion on possible future research and applications. I recommend revising the entire paper by making its aim and applicability more clear and consistent throughout the manuscript.

The experimental methods should be better described. How was a “crushing” defined in the present study? Did they result in piglets death or major injuries? If so, how were majour injuries defined? How many crushing events (and therefore calls) were observed and analyzed during the study? This last information in crucial to teh scientific soundness of the worls and is never stated. It should be repeated in every graph and table.

Another concern I have is the extremely limited literature review. Are the calls recorded in the present experiment comparable to calls recorded by others? How do they differ, if so, from other calls of pain (for example during castration)? Also, you state that calls are specific for crushing events. I'm not a "call expert", but I would assume that these calls might be an indication of extreme degree of pain and distress, and not a specific indicator of crushing.

I believe the manuscript should be better focused and shortened , in particular by widening the literary review in order to give readers a more comprehensive view.

I also attached a pdf file with more specific comments.

Author Response

Please see attached word document.

Round 2

Reviewer 1 Report

I believe the authors addressed most comments and have improved the manuscript. Critical for me was the addition of more detail in the methods, sample size and interpretation of the results.

I only have some minor comments/notes to make sure I understand correctly.

In regards to sample size, I understand now 10 sows - 1 crushing event per sow = 10 crushed piglets. Those 10 crushed piglets produced a total of 631 crushed calls (more than 1 call within a vocalization recording, average 60+) - average 62.9 calls per piglet. Note in the results L126-127 it says calls per sow, I think intuitively it will make more sense to write 'calls per piglet' as those are the ones producing the calls and not the sow. 

In the figures, the authors are now providing n = 10 piglets but I wonder if it would be even more clear to mention the 10 piglet and the number of calls. On that note, I realized where one of my points of initial confusion came from. The authors on occasion use the word 'vocalization' which I associated with one recording for one piglet and then having more calls within one vocalization. Reading the revised manuscript now, I believe that the authors are using the word vocalization and calls interchangeably and perhaps it can avoid confusion by naming all of them calls?

Author Response

Thank you for the feedback. All three edits have been modified in the document (total call has been added, title of figures now say 'per piglet', and calls has been substituted for 'vocalizations' throughout the document to clarify the confusion). 

Author Response

Thank you for the feedback.